# Patterns of Socioeconomic Inequities in SDGs Relating to Children’s Well-Being in Thailand and Policy Implications

**DOI:** 10.3390/ijerph192013626

**Published:** 2022-10-20

**Authors:** Orratai Waleewong, Khanuengnij Yueayai

**Affiliations:** 1The International Health Policy Program, Ministry of Public Health, Nonthaburi 11000, Thailand; 2Office of Disease Prevention and Control 9, Nakhon Ratchasima 30000, Thailand

**Keywords:** child health, women’s health, disparities, Thailand, MICS, SDGs, child flourishing index

## Abstract

Thailand faces many wealth inequities and child health-related problems. This study aimed to describe Thai child health and determine socioeconomic inequities following the child flourishing index, a tool used to measure children’s wellbeing based on the key relevant Sustainable Development Goals. The data from Thailand Multiple Indicator Cluster Survey 2019 were used to examine five indicators where Thailand had not yet achieved good results. The association of socioeconomic status with the five outcomes was explored using logistic regressions, comparing pseudo R-squared, and population attributable fraction analyses. Household wealth, urbanization, education, and primary language were significantly associated with Thai child health. Over 10% of children under 5 years were stunted and had a low birth weight. Fourteen percent of teenage girls had already become mothers. Living in poor households and rural areas, having a head-of-household who was non-Thai speaking, non-Buddhist, and had a low education were identified as risk factors for children with undernutrition status and low birth weight. However, having a head-of-household who spoke a non-Thai language was a protective factor against teenage mothers and having early marriages. Households with better economic status and education provided significant benefits for children and women’s health. The result of this study calls for public policies and multisectoral actions in the wider social and economic spheres that address the social determinants that span across lives and generations. Furthermore, specific social protection programs should be designed to be accessible by these most vulnerable and disadvantaged people.

## 1. Introduction

Despite some progress in reducing the mortality of children under 5 years old in addition to the reduction of neonatal and maternal mortality since the adoption of the Millennium Development Goals in 2005 [1,2], many aspects of children’s health and well-being still needs to be improved; inequalities remain significant, especially in low- and middle-income countries (LMICs). The number of maternal deaths was estimated to be 211 per 100,000 live births globally in 2017, of which around 20% had occurred in southern Asia. Neonatal deaths were estimated at 2.4 million in 2019. The prevalence of low birth-weight babies in 2015 was 14.60%, of which 91% had occurred in LMICs. A quarter of the children younger than 5 years old are stunted in about one-third of countries. Even though the prevalence of child marriage is decreasing on a global scale, the adolescent birth rate remains high, at 41 per 1000 adolescent girls in 2020 [3].

The Sustainable Development Goals (SDGs), which were developed to protect the planet from a dangerous and uncertain future and deliver secure, fair, and healthy lives for future generations [4], have been a platform for re-defining children’s well-being by placing children at the center of its endeavor. Investing in children’s health is irrefutably beneficial and has high benefit–cost ratios. To track the progress toward the SDGs relating to children’s well-being such as health, education, and nutrition, “A child flourishing index (CFI)”, was developed by the WHO–UNICEF–Lancet Commission in 2020. This new measure calls for stronger accountability and multisectoral actions for child health and well-being across the course of life [5]. Meanwhile, “leaving no one behind”, an overarching principle of the SDGs, reinforces efforts in combating child health inequities within and amongst countries. [6]. Thailand is one of the countries that faces many child health challenges, and the country was ranked the third most wealth-unequal country in the world in 2018 [7]. Even though the neonatal death rate is low (below 1%) in Thailand [8], children are not thriving. The thrive element in Thailand’s CFI scored 0.7 out of 1.0 in Thailand, which is lower than some LMICs. Poorer countries with lower CFI scores tend to have greater economic inequality [5]. This contradiction between the low neonatal death and the low levels of Thai children thriving must be explored in order to address the problem. A few studies have investigated children’s well-being in the context of the SDGs in Thailand [9,10], but there is lack of studies that comprehensively illustrate whether Thai children are receiving what they need to survive and thrive, as well as studies that illustrate the inequity challenges. The Thailand Multiple Indicator Cluster Survey (MICS), undertaken by the National Statistic Office following an international household survey program developed by [11], is the best available database, having national representativeness and providing rigorous data on the socioeconomic well-being of women, children, and households, along with matching with the CFI framework the best. This study aimed to describe the CFI indicator using the Thailand MICS 2019 data to determine the disparities in the health outcomes among specific socioeconomic groups of Thai children and their respective households. A better understanding of these situation can inform policy and establish a call for action in promoting Thai children’s wellbeing while addressing the barriers in achieving the SDGs.

## 2. Materials and Methods

### 2.1. Data Source and Study Design

Secondary data analyses were performed using the data from the Thailand MICS 2019 while following the CFI indicators, which measure the foundational conditions for children aged 0–18 years needed to survive and thrive. The CFI was constructed from 17 SDG indicators; 8 indicators for thriving (under-five mortality rate, maternal mortality ratio, third dose of diphtheria-tetanus-pertussis vaccine (DTP3) coverage, birth attended by skilled health personnel, prevalence of unsafe or unimproved sanitation, deaths due to road injuries for ages 0–19 years, children living in households in extreme poverty, children living below the national poverty line) and 9 indicators for surviving (expected years of school by age 20–24 years, harmonized mean test scores for age 15–19 years, suicide mortality rate for ages 15–24 years, prevalence of stunting among children under 5, children born with a low birth weight, adolescent birth rate for girls 10–19 years, proportion of women aged 20–24 years who were married or in union before age 15, proportion of young women and men aged 18–29 years that experienced sexual violence by age 18, proportion of ever-partnered women aged 15–49 years that were subjected to violence by a partner in the past 12 months). Details of the CFI calculations can be found elsewhere [5]. Due to several limitations, the CFI is considered as the process of raising awareness of the need to measure and promote conditions fundamental to child wellbeing at this early phase of implementation. The Thailand MICS 2019 collected data through a trained interviewer team, consisting of 2–4 interviewers, a supervisor, and a translator or non-Thai household interview, using computer-assisted personal interviewing. The data collection application was based on the Census and Survey Processing System software, version 6.3. Data were recorded on a laptop computer from the sample of stratified two-stage sampling among men (aged 15–49) and the following groups of interest: women aged 15–49 years, children under 5 years, and children aged 5–14 years; the completeness of the answers for these groups were 92.0%, 94.0%, and 93.8%, respectively [11].

### 2.2. Variables

The MICS 2019 data could be generated as proxies of 8 out of the 17 CFI indicators, which were DTP3 coverage, birth attended by skilled health personnel, prevalence of unsafe or unimproved sanitation, children living below the national poverty line, prevalence of stunting among children under 5 years, children born with a low birth weight, adolescent birth rate for girls 10–19 years, and proportion of women aged 20–24 years who were married or in union before age 15 (Appendix A). All unspecified or blank answers of both the dependent and independent variables were identified as invalid and were excluded from the analysis.

The independent variables were socioeconomic factors: household income (1st quintile as the reference group versus 2nd, 3rd, 4th, and 5th quintile); residential areas (urban as the reference group versus rural); residential region (Bangkok as the reference group versus central, north, northeast, and south); language (Thai as the reference group versus non-Thai languages, which were identified as English, Chinese, Burmese, Malaysian/Jawi; ethnic languages included Khmer/Kuy, Lao, Karen, Hmong, Lahu, Mon, Lawa, Akha, Nyeu, and Shan) and religion of the head-of-household (HOH) (Buddhism as the reference group versus Islam, Christianity, Others, and no religion); the HOH, maternal, and individual education levels of family members (kindergarten or none as the reference group versus primary (grade 1–6), lower secondary (grade 7–9), upper secondary (grade 10–12 or equal), and higher (higher than grade 12)); and sex (male as the reference group versus female).

The dependent variables included 8 main outcomes following the CFI indicators calculation [5]: births attended by skilled health personnel (attended by skilled health personnel defined as “1” versus attended by others defined as “0”), DTP3 coverage (received 3 doses of the DTP vaccine defined as “1” versus received less than 3 doses defined as “0”), using unsafe or unimproved sanitation (used unsafe or unimproved sanitation defined as “1” versus used improved sanitation defined as “0”); data for those living below the national poverty line data were not available, which led to us applying households being in extreme poverty instead (being the 1st quintile household income defined as “1” versus other quintiles defined as “0”), stunting (being stunted defined as “1” versus normal height for age defined as “0”), low birth weight (born with a low birth weight defined as “1” versus non-low birth weight defined as “0”), adolescent birth (giving birth at 15–19 years (adolescent birth) defined as “1” versus non-adolescent birth defined as “0”), and women aged 20–24 years who were married or in union before age 15 years (married or in union before age 15 years (early married) defined as “1” versus non-early married defined as “0”) (Appendix A).

### 2.3. Statistical Analysis

The eight indicators, which were selected based on the matched variables between the CFI and MICS, were described using percentages (Appendix A). Further exploration was conducted in 5 indicators where Thailand does not yet have a good score: children living below the national poverty line, prevalence of stunting among children under 5 years old, children born with a low birth weight, adolescent birth rate for girls 10–19 years, and proportion of women aged 20–24 years who were married or in union before age 15 years. A Pearson’s chi-squared test with Bonferroni correction analysis was used to determine the different proportion of each characteristic among a group of people who had (defined as “1”) and who did not have the interested outcomes (defined as “0”). The odds ratios (ORs), Adjusted odds ratios (AORs), 95% confidence intervals (CIs), and *p*-values for the associations between each health outcome and the socioeconomic status were estimated using a logistic regression analysis. The household poverty, residential area and region, HOH language, HOH religion, HOH education, women education, parity, number of household members, and number of children and adolescents in the household variables were adjusted in a multivariate analysis. The pseudo R-squared measure was compared between the full model and the full model without a variable of interest to see the effect of each variable/socioeconomic status on the outcome variance. Moreover, the population attributable fraction (PAF) was calculated to estimate the proportional reduction in cases where the exposure to risk factors was eliminated using sequential and average AF, as described previously [12,13]. All tests were unweighted and two-sided with the alpha value set at 0.05; the tests were performed using Stata version 14.2.

### 2.4. Ethics Approval

This study received a review exemption from the Institute for the Development of Human Research Protections, Thailand (COE No. IHRP2021001/IHRP and No. 198-2564).

## 3. Results

### 3.1. Overall of Thailand’s Progress in CFI

Among the eight CFI indicators in this study, Thailand already achieved the highest score estimation for three of the indicators, including births attended by skilled health personnel (99.8%), DTP3 coverage (95.3%), and proportion of populations using unsafe or unimproved sanitation (2.3%), all of which used improved toilet types but shared their facility with others who were not members of their household. The proportion of non-Thai households using unimproved sanitation was much higher than those using improved sanitation (21.4% vs. 10.0%). The remaining five CFI indicators still need improving, including children living in poverty, prevalence of stunting among children under 5 years, children born with a low birth weight, adolescent birth rate, and proportion of women aged 20–24 years who were married or in union before age 15.

#### 3.1.1. Socioeconomic Differences in Household Poverty

A total of 28.4% (10,054/35,393) households belonged to the poorest wealth index quintile. The poorest households were significantly higher in rural rather than urban areas (*p* < 0.001). Households in the southern region had a lower chance of being in the poorest quintile compared with households that resided in Bangkok (AOR = 0.82, 95% CI: 0.71, 0.94), whereas households in other regions had a significantly higher risk of poverty. Non-Thai speaking HOHs (58.5% of them spoke ethnic languages and 39.5% spoke Malaysian/Jawi language) were more frequently found among the poorest (16.5%) rather than wealthier households (6.8%). Therefore, the risk of being among the poorest was higher in households where the HOH spoke non-Thai languages (AOR = 2.90, 95% CI 2.61, 3.22). Compared with households with a Buddhist HOH, households with a Christian HOH (AOR = 1.76, 95% CI 1.36, 2.28) and HOH with no religion (AOR = 5.64, 95% CI 1.40, 22.61) had a higher chance of being in the lowest wealth index quintile. Furthermore, having an HOH with higher education levels was significantly associated with a lower likelihood of being in extreme poverty (*p* < 0.001), as noted in Table 1. Around 38.6% of the poorest households were attributed to the people living outside of Bangkok, whereas 19.6% were attributed to those who lived in rural areas and 10.2% were attributed to those who had an HOH with the lowest school level attended. The HOH education explained 7.8% of the variance of being in the poorest household, which was followed by the difference of pseudo R-squared values (Table 1).

#### 3.1.2. Socioeconomic Differences in Growth and Nutrition among Thai Children (SDG 2.2.1)

Data were available for a total of 13,649 children under 5 years with height according to their age. A group of 1733 (12.7%) children were stunted. Having a better household financial status protected children from being stunted (*p* < 0.001). Among those stunted children, 16.2% of the HOHs spoke non-Thai languages (of which 61.8% spoke Malaysian/Jawi languages and 37.1% spoke ethnic languages); this was higher than among the non-stunted children (9.7%). Therefore, the risk of being stunted among children with HOHs who spoke non-Thai languages was higher (AOR = 1.21, 95%CI 1.01, 1.46). Compared with children with a Buddhist HOH, children with an Islamic HOH had a higher chance of being stunted (AOR = 1.39, 95%CI 1.13, 1.71). Better HOH education was also shown to be a protective factor, as shown in Table 2.

A total of 11,510 children under 5 years had available documented birth weight data. A group of 1103 (9.6%) children had a low birth weight. Overall, having a better household wealth prevented children from having a low birth weight (*p* = 0.022). Being female increased the risk of having a low birth weight compared with being male (AOR = 1.23, 95% CI 1.08, 1.39), as noted in Table 3. Socioeconomic status did not attribute much to the result of being stunted by the models that we used to calculate the PAF and pseudo R-squared values. However, living in a Thai household was attributed the most to having a low birth weight, accounting for 21.4% of the variation, followed by being female, which accounted for about 10.0% (Table 3).

#### 3.1.3. Socioeconomic Differences among Thai Female Adolescents (SDG 3.7.2)

From a total of 2847 women aged 15–19 years, 389 (13.7%) already had at least one child. Compared with women who attended kindergarten school or lower, adolescent women who attended upper-secondary school (AOR = 0.09, 95% CI 0.02, 0.38) and higher than upper-secondary school (AOR = 0.01, 95% CI 0.00, 0.15) were less likely to have a child early, as noted in Table 4.

From a total of 2953 women aged 20–24 years old, 201 (6.8%) had married or were in union before being 15 years old. The overall effect of regions significantly predicted the probability of early marriage (*p* = 0.022); however, there was no significant difference in chances of early marriage in any region compared with Bangkok. The significantly protective factor was education, as women who attended upper-secondary school (AOR = 0.21, 95% CI 0.06, 0.78) and higher than upper-secondary school (AOR = 0.02, 95% CI 0.00, 0.12) were less likely to be in union or married early compared with women who attended kindergarten school or lower, as shown in Table 5. Around 60.4% of the women that had a child at an adolescent age was attributable to subjects who lived with an HOH who attended higher than kindergarten school level, and 24.8% was attributed to living in a Thai household (Table 4). Around 49.3% of women being married early was attributed to them living in a Thai household. The lowest level of school attendance explained 13.0% of the variance of having a child at an adolescent age and explained 6.6% of the variance of early marriage, which was much higher than other factors (Table 5).

## 4. Discussion

This study highlighted the persistence and widespread prevalence of socioeconomic inequities and their effects in Thailand. Overall, household poverty, residential areas/regions, education, and primary language of the HOH were key determinants of the health and well-being of Thai children. A small proportion of children lived in households using unimproved sanitation, but due to the large sample size this proportion represented thousands of households. This basic facility was an important burden on the health of both children and adults [14,15,16]. Having an HOH with a better education, living in Bangkok (except in the southern region, which was the richest region but had the highest disparities among the poorest and richest groups [17]), and living in urban areas were benefits to child health and well-being (Appendix A). However, those economic and education opportunities were still centralized in the capital and urban areas [18]. Around 1.8 million poor children potentially missed out on education due to it being unaffordable. The education opportunity among the poorest children was 20 times lower than the richest children. Furthermore, the development of schools in rural areas was two years more delayed than schools in urban areas [19]. Having an ethnic, Christian, or non-religious HOH were risk factors for poverty. Furthermore, previous studies have shown a positive relation between ethnic minorities and poverty [20]. Furthermore, religion is linked to poverty by limited participation in some jobs and communities due to beliefs and attitudes [21].

The prevalence of stunting among children under 5 years was 12.7%, which is slightly lower than the global SDG 2019 report at 13.4% [22]. Households being wealthy and having an HOH with a higher education had a greater chance of obtaining better nutrition for their children; furthermore, these children were less likely to have been stunted, as shown in previous studies [23,24,25]. Non-Thai households and having an Islamic instead of Buddhist HOH were risk factors for stunting. Non-Thai households were related to lower income and the use of unimproved sanitation, as mentioned above. Using unimproved sanitation was also associated with a higher risk of stunting [26,27]. The risk of stunting among Muslim children was also found in other countries in South Asia [28,29].

The amount of children born with a low birth weight was 9.6% (8.7% when weighted for the national population [11], which is slightly different from World Bank figures in 2015 (10.5%)) [30]. Living in a poor household and being female were risk factors. Previous studies have also shown that female children were more likely to have lower birth weights than male children [31,32], and poverty is significantly associated with the prevalence of a low birth weight [33,34]. The models we used did not predict the probability of having stunting and low birth weight well, and other significant related factors [35,36,37] could not be included.

The adolescent birth rate was 136.6 per 1000 women aged 15–19 years (54/1000 when weighted for the national population [11]), which is higher than the results obtained from the previous version of the Thailand MICS report, which noted 51 births per 1000 women aged 15–19 years [38]. The variation of unweighted and weighted proportion made it possible to reflect that the sample was not representative. A higher education in the girls was a protective factor; these results were similar to studies in other countries [39,40]. A risk factor was having an HOH who had a higher education level. An even higher education levels of HOH should be a protective factor. However, adolescents who had early births often lived with a household that had the ability to provide financial support [41]. A possible explanation is that an HOH with a higher education is associated with a better household income. Poor education around reproductive health and birth control were significantly associated with teenage pregnancy [42,43]. Therefore, in Thailand, access to reproductive health community services should be proactively promoted, especially in school settings.

The proportion of women aged 20–24 years who were married or in union before being 15 years old was observed to be 6.8% (3.0% when weighted for the national population [11]), whereas the World Bank data in 2019 reported 3.0% [44]. Based on our results, living with non-Thai households and girls having a higher education were protective factors; the education factor being a protective factor is similar to previous studies in Thailand and other countries [45,46]. A study in Chiang Mai, Thailand found that Thai adolescents were more likely to engage in sexual intercourse at an early age compared with ethnic minority adolescents [47]. This could imply that Thai women are more likely to be married early than ethnic women.

We attempted to control for confounders by adjusting for covariates, but the outcomes had multiple factors involved. These results are possibly confounded by other factors not available in the MICS 2019 data. However, our results are consistent with previous studies. Therefore, the subgroups of the population that indicated a higher probability of poor health outcomes should be considered a priority for further investigations and interventions.

In summary, the revealed socioeconomic inequities in the SDGs that relate to children’s wellbeing call for public policies and multisectoral actions in the wider social and economic spheres that address the social determinants that span across lives and generations, which is similar to other countries around the world [48]. At macro-level, policies that encourage urban decentralization should be pursued to distribute economic and education opportunities to those living in rural areas. At the community level, access to infrastructure and basic facilities such as water and sanitation should be proactively improved in disadvantaged areas. The improvement of the level and distribution of social protection (such as school feeding) and policies that improve the affordability of healthy foods could contribute to better child nutrition status, particularly among the poor and among households having an HOH with a low education. Furthermore, we found health and economic vulnerabilities among these marginalized households. Thus, specific social protection programs should be designed to be accessible by these most vulnerable and disadvantaged people. Further studies should explore how to overcome barriers to achieve better health outcomes among this group through the implementation of effective interventions. More importantly, at the societal level, policies oriented toward social inclusion and women’s rights and gender equality should be promoted as part of development toward an equitable and sustainable society for future generations.

## 5. Conclusions

Resonating with the SDGs’ overarching principle of leaving no one behind, the socioeconomic inequities in children’s wellbeing in Thailand is persistent and widespread. Household wealth, urbanization, education level, and primary language of a head-of-household were significantly associated with Thai children’s health and wellbeing. The results of this study call for public policies and multisectoral actions in the wider social and economic spheres that address the social determinants that span across lives and generations.

## Figures and Tables

**Table 1 ijerph-19-13626-t001:** The proportion of households in extreme poverty by socioeconomic status (SES), association between being in an extreme poverty household and socioeconomic status, and effect of each SES factor as illustrated by the PAF and different pseudo-R-squared values.

Category		Other Wealth Index Quintiles (0)	The Lowest Wealth Index Quintile (1)	Chi^2^ (*p*-Value)	Univariate		Multivariate		Population Attributable Fraction: PAF	Different Pseudo R^2^
No. (Column%)		No. (Column%)	No. (Column%)		OR (95%CI)	*p* Value	AOR (95%CI)	*p* Value		
Poverty status (n = 35,393)										
	Poorest: 10,054 (28.40%)									
	Poor: 7953 (22.47%)									
	Middle: 6861 (19.39%)									
	Rich: 5847 (16.52%)									
	Richest: 4678 (13.22%)									
Residential area (n = 35,393)										
	Urban (municipal): 14,146 (39.97%)	11,305 (44.62%)	2841 (28.26%)	802.70 (<0.001) *	Reference				Reference	0.0026
	Rural (non-municipal): 21,247 (60.03%)	14,034 (55.38%)	7213 (71.74%)		2.04 (1.94, 2.15)	<0.001 *	1.38 (1.30, 1.46)	<0.001 *	0.1960	
Residential region (n = 35,391)						<0.001 *		<0.001 *		
	Bangkok (a): 3436 (9.71%)	3014 (11.89%)	422 (4.20%)	892.99 (<0.001) * all are distinct **	Reference				Reference	0.0280
	Central (b): 8768 (24.77%)	6647 (26.24%)	2121 (21.10%)		2.27 (2.03, 2.55)	<0.001 *	1.49 (1.32, 1.69)	<0.001 *	0.3862	
	Northern (c): 5452 (15.4%)	3860 (15.23%)	1592 (15.83%)		2.94 (2.61, 3.31)	<0.001 *	1.28 (1.12, 1.46)	<0.001 *		
	Northeastern (d): 10,734 (30.33%)	6280 (24.78%)	4454 (44.30%)		5.06 (4.54, 5.64)	<0.001 *	3.11 (2.75, 3.52)	<0.001 *		
	Southern (e): 7003 (19.79%)	5538 (21.86%)	1465 (14.57%)		1.88 (1.68, 2.12)	<0.001 *	0.82 (0.71, 0.94)	0.005 *		
Head of household language (n = 35,393)										
	Thai: 32,018 (90.46%)	23,627 (93.24%)	8391 (83.46%)	798.80 (<0.001) *	Reference				Reference	0.0097
	Non-Thai: 3375 (9.54%)	1712 (6.76%)	1663 (16.54%)		2.73 (2.54, 2.93)	<0.001 *	2.90 (2.61, 3.22)	<0.001 *	0.1081	
Head of household religion (n = 35,393)						<0.001 *		<0.001 *		
	Buddhism (a): 31,735 (89.67%)	22,836 (90.13%)	8899 (88.51%)	151.02 (<0.001) * a and b differ from d **	Reference				Reference	0.0007
	Islam (b): 3296 (9.31%)	2345 (9.26%)	951 (9.46%)		1.04 (0.96, 1.12)	0.324	1.13 (0.99, 1.29)	0.062	0.0225	
	Christianity (c): 336 (0.95%)	147 (0.58%)	189 (1.88%)		3.29 (2.65, 4.09)	<0.001 *	1.76 (1.36, 2.28)	<0.001 *		
	Others (d): 9 (0.03%)	6 (0.02%)	3 (0.03%)		1.28 (0.32, 5.13)	0.725	2.38 (0.97, 5.86)	0.058		
	No religion (e): 15 (0.04%)	3 (0.01%)	12 (0.12%)		10.2 (2.89, 36.3)	<0.001 *	5.64 (1.40, 22.61)	0.015 *		
Head of household education (n = 35,367)						<0.001 *		<0.001 *		
	Kindergarten or none (a): 2254 (6.36%)	883 (3.48%)	1371 (13.65%)	3600.00 (<0.001) * all are distinct **	Reference				0.1024	0.0778
	Primary (b): 20,950 (59.24%)	13,670 (53.98%)	7280 (72.51%)		0.34 (0.31, 0.37)	<0.001 *	0.38 (0.34, 0.42)	<0.001 *	Reference	
	Lower Secondary (c): 3540 (10.01%)	2822 (11.14%)	718 (7.15%)		0.16 (0.14, 0.18)	<0.001 *	0.18 (0.16, 0.21)	<0.001 *		
	Upper Secondary (d): 4073 (11.52%)	3540 (13.98%)	533 (5.31%)		0.09 (0.08, 0.10)	<0.001 *	0.11 (0.09, 0.12)	<0.001 *		
	Higher (e): 4550 (12.87%)	4411 (17.42%)	139 (1.38%)		0.02 (0.01, 0.02)	<0.001 *	0.02 (0.02, 0.03)	<0.001 *		
Head of household sex (n = 35,392)										
	Male: 20,695 (58.47%)	14,962 (59.05%)	5733 (57.02%)	12.19 (<0.001) *	Reference				0.0648	0.0005
	Female: 14,697 (41.53%)	10,376 (40.95%)	4321 (42.98%)		1.08 (1.03, 1.13)	<0.001 *	0.89 (0.84, 0.94)	<0.001 *	Reference	

Different pseudo R^2^: comparison of the full model (pseudo R^2^ = 0.1871) and the full model without an interesting factor; the full model adjusted for the residential area and region, HOH language, HOH religion, HOH education, HOH sex, number of household members, and number of children and adolescents in the household; * statistically significant at alpha = 0.05; ** statistically significant at Bonferroni correction alpha = 0.0005.

**Table 2 ijerph-19-13626-t002:** The proportion of stunting among children under 5 years by socioeconomic status (SES), association between being stunted and socioeconomic status, and effect of each SES as illustrated by the PAF and different pseudo-R-squared values.

Category		Normal Height for Age (0)	Stunting (1)	Chi^2^ (*p*-Value)	Univariate		Multivariate		Population Attributable Fraction: PAF	Different Pseudo R^2^
No. (Column%)		No. (Column%)	No. (Column%)		OR (95%CI)	*p* Value	AOR (95%CI)	*p* Value		
Poverty status (n = 13,649)						<0.001 *		<0.001 *		
	Poorest (a): 3338 (24.46%)	2805 (23.54%)	533 (30.76%)	57.13 (<0.001) * a differs from c,d,e **	Reference				0.0700	0.0021
	Poor (b): 3128 (22.91%)	2717 (22.80%)	411 (23.72%)		0.79 (0.69, 0.91)	0.001 *	0.83 (0.71, 0.95)	0.008 *	Reference	
	Middle (c): 2888 (21.16%)	2539 (21.31%)	349 (20.13%)		0.72 (0.62, 0.83)	<0.001 *	0.76 (0.65, 0.88)	<0.001 *		
	Rich (d): 2489 (18.24%)	2237 (18.77%)	252 (14.54%)		0.59 (0.50, 0.69)	<0.001 *	0.68 (0.57, 0.81)	<0.001 *		
	Richest (e): 1806 (13.23%)	1618 (13.58%)	188 (10.85%)		0.61 (0.51, 0.72)	<0.001 *	0.78 (0.63, 0.97)	0.026 *		
Residential area (n = 13,649)										
	Urban (municipal): 4674 (34.24%)	4118 (34.56%)	556 (32.08%)	4.12 (0.042) *	Reference				0.0038	0.0000
	Rural (non-municipal): 8975 (65.76%)	7798 (65.44%)	1177 (67.92%)		1.11 (1.00, 1.24)	0.043 *	0.99 (0.88, 1.11)	0.843	Reference	
Residential region (n = 13,649)						<0.001 *		<0.001 *		
	Bangkok (a): 677 (4.96%)	593 (4.98%)	84 (4.85%)	54.79 (<0.001) * b differs from c,d,e **	Reference				0.0074	0.0027
	Central (b): 3562 (26.10%)	3222 (27.04%)	340 (19.62%)		0.74 (0.57, 0.96)	0.023 *	0.68 (0.52, 0.90)	0.005*	Reference	
	Northern (c): 2061 (15.10%)	1774 (14.89%)	287 (16.56%)		1.14 (0.88, 1.48)	0.317	1.00 (0.76, 1.33)	0.977		
	Northeastern (d): 4372 (32.03%)	3809 (31.97%)	563 (32.49%)		1.04 (0.81, 1.33)	0.734	0.92 (0.70, 1.20)	0.532		
	Southern (e): 2977 (21.81%)	2518 (21.13%)	459 (26.49%)		1.28 (1.00, 1.65)	0.047 *	0.95 (0.72, 1.26)	0.726		
Household leader’s language (n = 13,649)										
	Thai: 12,213 (89.48%)	10,760 (90.30%)	1453 (83.84%)	66.98 (<0.001) *	Reference				Reference	0.0004
	Non-Thai: 1436 (10.52%)	1156 (9.70%)	280 (16.16%)		1.79 (1.55, 2.06)	<0.001 *	1.21 (1.01, 1.46)	0.043 *	0.0282	
Household leader’s religion (n = 13,649)						<0.001 *		0.007 *		
	Buddhism (a): 11,970 (87.71%)	10,555 (88.59%)	1415 (81.65%)	68.32 (<0.001) * a differs from b **	Reference				Reference	0.0009
	Islam (b): 1526 (11.18%)	1236 (10.37%)	290 (16.73%)		1.75 (1.52, 2.01)	<0.001 *	1.39 (1.13, 1.71)	0.002 *	0.0484	
	Christianity (c): 149 (1.09%)	121 (1.02%)	28 (1.62%)		1.72 (1.14, 2.61)	0.010 *	1.25 (0.81, 1.92)	0.310		
	Others (d): 0 (0.00%)	0 (0.00%)	0 (0.00%)							
	No religion (e): 2 (0.01%)	2 (0.02%)	0 (0.00%)							
Household leader’s education (n = 13,649)						<0.001 *		0.097		
	Kindergarten or none (a): 737 (5.41%)	602 (5.06%)	135 (7.79%)	36.91 (<0.001) * a differs from b,d,e and b differs from e **	Reference				0.0144	0.0008
	Primary (b): 8080 (59.25%)	7044 (59.16%)	1036 (59.82%)		0.65 (0.53, 0.79)	<0.001 *	0.83 (0.65, 1.05)	0.128	Reference	
	Lower Secondary (c): 1574 (11.54%)	1363 (11.45%)	211 (12.18%)		0.69 (0.54, 0.87)	0.002 *	0.83 (0.63, 1.10)	0.194		
	Upper Secondary (d): 1738 (12.74%)	1531 (12.86%)	207 (11.95%)		0.60 (0.47, 0.76)	<0.001 *	0.74 (0.56, 0.98)	0.035 *		
	Higher (e): 1509 (11.06%)	1366 (11.47%)	143 (8.26%)		0.46 (0.36, 0.60)	<0.001 *	0.66 (0.49, 0.91)	0.010*		
Maternal education (n = 13,649)						<0.001 *		0.290		
	Kindergarten or none (a): 463 (3.39%)	386 (3.24%)	77 (4.44%)	25.97 (<0.001) * a and c differ from e **	Reference				Reference	0.0005
	Primary (b): 4221 (30.94%)	3671 (30.82%)	550 (31.74%)		0.75 (0.57, 0.97)	0.031 *	0.98 (0.73, 1.35)	0.946	0.0388	
	Lower Secondary (c): 2684 (19.67%)	2307 (19.37%)	377 (21.75%)		0.81 (0.62, 1.07)	0.144	1.14 (0.83, 1.56)	0.428		
	Upper Secondary (d): 3104 (22.75%)	2706 (22.72%)	398 (22.97%)		0.73 (0.56, 0.96)	0.025 *	1.09 (0.79, 1.50)	0.597		
	Higher (e): 3172 (23.25%)	2841 (23.85%)	331 (19.10%)		0.58 (0.44, 0.76)	<0.001 *	0.97 (0.69, 1.35)	0.842		
Sex (n = 13,649)										
	Male: 7012 (51.37%)	6094 (51.14%)	918 (52.97%)	2.03 (0.154)	Reference				0.0383	0.0002
	Female: 6637 (48.63%)	5822 (48.86%)	815 (47.03%)		0.92 (0.84, 1.02)	0.154			Reference	

Different pseudo R^2^: comparison of the full model (pseudo R^2^ = 0.0156) and the full model without an interesting factor; the full model adjusted for household poverty, residential area and region, HOH language, HOH religion, HOH education, maternal education, sex, number of household members, number of children and adolescents in the household; * statistically significant at alpha = 0.05; ** statistically significant at Bonferroni correction alpha = 0.0005.

**Table 3 ijerph-19-13626-t003:** The proportion of children born with low birth weight by socioeconomic status (SES), association between having low birth weight and socioeconomic status, and effect of each SES as illustrated by the PAF and different pseudo-R-squared values.

Category		Non-Low Birth Weight (0)	Low Birth Weight (1)	Chi^2^ (*p* Value)	Univariate		Multivariate		Population Attributable Fraction: PAF	Different Pseudo R^2^
No. (Column%)		No. (Column%)	No. (Column%)		OR (95%CI)	*p* Value	AOR (95%CI)	*p* Value		
Poverty status (n = 11,510)						0.002 *		0.022 *		
	Poorest (a): 2907 (25.26%)	2589 (24.88%)	318 (28.83%)	16.49 (0.002) * a differs from e **	Reference				0.0437	0.0016
	Poor (b): 2692 (23.39%)	2412 (23.18%)	280 (25.39%)		0.94 (0.79, 1.12)	0.518	0.95 (0.80, 1.13)	0.538	Reference	
	Middle (c): 2438 (21.18%)	2225 (21.38%)	213 (19.31%)		0.77 (0.64, 0.93)	0.007 *	0.78 (0.64, 0.4)	0.011 *		
	Rich (d): 2070 (17.98%)	1903 (18.29%)	167 (15.14%)		0.71 (0.58, 0.86)	0.001 *	0.74 (0.59, 0.92)	0.006 *		
	Richest (e): 1403 (12.19%)	1278 (12.28%)	125 (11.33%)		0.79 (0.64, 0.99)	0.040 *	0.88 (0.67, 1.15)	0.348		
Residential area (n = 11,510)										
	Urban (municipal): 3789 (32.92%)	3442 (33.07%)	347 (31.46%)	1.18 (0.278)	Reference				Reference	0.0000
	Rural (non-municipal): 7721 (67.08%)	6965 (66.93%)	756 (68.54%)		1.07 (0.94, 1.23)	0.276			0.0322	
Residential region (n = 11,510)						0.484				
	Bangkok (a): 458 (3.98%)	415 (3.99%)	43 (3.90%)	3.48 (0.481)	Reference				0.0012	0.0003
	Central (b): 2936 (25.51%)	2674 (25.69%)	262 (23.75%)		0.94 (0.67, 1.32)	0.745			Reference	
	Northern (c): 1770 (15.38%)	1606 (15.43%)	164 (14.87%)		0.98 (0.69, 1.40)	0.936				
	Northeastern (d): 3918 (34.04%)	3518 (33.80%)	400 (36.26%)		1.09 (0.78, 1.52)	0.584				
	Southern (e): 2428 (21.09%)	2194 (21.08%)	234 (21.21%)		1.02 (0.73, 1.44)	0.870				
Household leader’s language (n = 11,510)										
	Thai: 10,361 (90.02%)	9361 (89.95%)	1000 (90.66%)	0.56 (0.453)	Reference				0.2135	0.0005
	Non-Thai: 1149 (9.98%)	1046 (10.05%)	103 (9.34%)		0.92 (0.74, 1.14)	0.455			Reference	
Household leader’s religion (n = 11,508)						0.445				
	Buddhism (a): 10,152 (88.22%)	9187 (88.29%)	965 (87.49%)	3.02 (0.388)	Reference				Reference	0.0003
	Islam (b): 1215 (10.56%)	1096 (10.53%)	119 (10.79%)		1.03 (0.84, 1.26)	0.744			0.0199	
	Christianity (c): 139 (1.21%)	120 (1.15%)	19 (1.72%)		1.49 (0.91, 2.43)	0.106				
	Others (d): 0 (0.00%)	0 (0.00%)	0 (0.00%)							
	No religion (e): 2 (0.02%)	2 (0.02%)	0 (0.00%)							
Household leader’s education (n = 11,502)						0.308				
	Kindergarten or none (a): 600 (5.22%)	535 (5.14%)	65 (9.50%)	4.85 (0.303)	Reference				0.0084	0.0002
	Primary (b): 65,958 (60.49%)	6218 (60.39%)	677 (61.43%)		0.88 (0.67, 1.16)	0.390			Reference	
	Lower Secondary (c): 1312 (11.41%)	1181 (11.36%)	131 (11.89%)		0.91 (0.66, 1.25)	0.578				
	Upper Secondary (d): 1443 (12.55%)	1310 (12.60%)	133 (12.07%)		0.83 (0.61, 1.14)	0.264				
	Higher (e): 1189 (10.34%)	1093 (10.51%)	96 (8.71%)		0.72 (0.51, 1.00)	0.056				
Maternal education (n = 11,507)						0.019 *		0.066		
	Kindergarten or none (a): 368 (3.20%)	333 (3.20%)	35 (3.17%)	13.18 (0.01) * b differs from d **	Reference				Reference 0.0854	0.0012
	Primary (b): 3620 (31.46%)	3254 (31.28%)	366 (33.18%)		1.06 (0.74, 1.54)	0.718	1.07 (0.71, 1.63)	0.727	0.0433	
	Lower Secondary (c): 2251 (19.56%)	2010 (19.32%)	241 (21.85%)		1.14 (0.78, 1.65)	0.491	1.12 (0.73, 1.70)	0.607		
	Upper Secondary (d): 2677 (23.26%)	2420 (23.26%)	257 (23.30%)		1.01 (0.69, 1.46)	0.956	1.01 (0.66, 1.54)	0.969		
	Higher (e): 2591 (22.52%)	2387 (22.94%)	204 (18.50%)		0.81 (0.55, 1.18)	0.280	0.80 (0.52, 1.25)	0.328		
Sex (n = 11,510)										
	Male: 5943 (51.63%)	5424 (52.12%)	519 (47.05%)	10.25 (0.001) *	Reference				Reference	0.0014
	Female: 5567 (48.37%)	4983 (47.88%)	584 (52.95%)		1.22 (1.08, 1.38)	0.001 *	1.23 (1.08, 1.39)	0.001 *	0.0999	

Different pseudo R^2^: comparison of the full model (pseudo R^2^ = 0.0092) and the full model without an interesting factor; the full model adjusted for household poverty, residential area and region, HOH language, HOH religion, HOH education, maternal education, gender, number of household members, number of children and adolescents in the household; * statistically significant at alpha = 0.05; ** statistically significant at Bonferroni correction alpha =0.0005.

**Table 4 ijerph-19-13626-t004:** The proportion of adolescent births by socioeconomic status (SES), association between adolescent birth and socioeconomic status, and effect of each SES as illustrated by the PAF and different pseudo-R-squared values.

Category		Non-Adolescent Birth (0)	Adolescent Birth (1)	Chi^2^ (*p* Value)	Univariate		Multivariate		Population Attributable Fraction: PAF	Different Pseudo R^2^
No. (Column%)		No. (Column%)	No. (Column%)		OR (95%CI)	*p* Value	AOR (95%CI)	*p* Value		
Poverty status (n = 2847)						<0.001 *		0.225		
	Poorest (a): 656 (23.04%)	520 (21.16%)	136 (34.96%)	46.05 (<0.001) * a differs from c,d,e **	Reference				Reference	0.0042
	Poor (b): 726 (25.50%)	621 (25.26%)	105 (26.99%)		0.64 (0.48, 0.85)	0.002 *	0.99 (0.61, 1.61)	0.965	0.0200	
	Middle (c): 605 (21.25%)	537 (21.85%)	68 (17.48%)		0.48 (0.35, 0.66)	<0.001 *	0.92 (0.52, 1.60)	0.759		
	Rich (d): 484 (17.00%)	434 (17.66%)	50 (12.85%)		0.44 (0.31, 0.62)	<0.001 *	1.76 (0.93, 3.31)	0.081		
	Richest (e): 376 (13.21%)	346 (14.07%)	30 (7.72%)		0.33 (0.21, 0.50)	<0.001 *	0.75 (0.35, 1.63)	0.466		
Residential area (n = 2847)										
	Urban (municipal): 1051 (36.92%)	921 (37.47%)	130 (33.42%)	2.37 (0.124)	Reference				Reference	0.0001
	Rural (non-municipal): 1796 (63.08%)	1537 (62.53%)	259 (66.58%)		1.19 (0.95, 1.49)	0.124			0.01164	
Residential region (n = 2847)						<0.001 *		0.517		
	Bangkok (a): 223 (7.83%)	203 (8.26%)	20 (5.14%)	25.12 (<0.001) * b differs from a,e **	Reference				Reference	0.0027
	Central (b): 746 (26.20%)	607 (24.69%)	139 (35.73%)		2.32 (1.41, 3.81)	0.001 *	1.32 (0.61, 2.83)	0.480	0.1067	
	Northern (c): 374 (13.14%)	322 (13.10%)	52 (13.37%)		1.63 (0.95, 2.82)	0.075	1.39 (0.56, 3.46)	0.485		
	Northeastern (d): 821 (28.84%)	717 (29.17%)	104 (26.74%)		1.47 (0.89, 2.43)	0.132	0.98 (0.43, 2.23)	0.957		
	Southern (e): 683 (23.99%)	609 (24.78%)	74 (19.02%)		1.23 (0.73, 2.07)	0.428	1.80 (0.30, 2.12)	0.653		
Household leader’s language (n = 2847)										
	Thai: 2461 (86.44%)	2103 (85.56%)	358 (92.03%)	12.01 (0.001) *	Reference				0.2475	0.0005
	Non-Thai: 386 (13.56%)	355 (14.44%)	31 (7.97%)		0.51 (0.34, 0.75)	0.001 *	0.73 (0.33, 1.61)	0.434	Reference	
Household leader’s religion (n = 2847)						0.002 *		0.748		
	Buddhism (a): 2380 (83.60%)	2031 (82.63%)	349 (89.72%)	21.68 (<0.001) * a differs from b **	Reference				0.2152	0.0000
	Islam (b): 438 (15.38%)	404 (16.43%)	34 (8.73%)		0.48 (0.33, 0.70)	<0.001 *	0.72 (0.26, 1.87)	0.499	Reference	
	Christianity (c): 28 (0.98%)	23 (0.94%)	5 (1.29%)		1.26 (0.47, 3.34)	0.636	0.69 (0.10, 4.56)	0.698		
	Others (d): 0 (0.00%)	0 (0.00%)	0 (0.00%)							
	No religion (e): 1 (0.04%)	0 (0.00%)	1 (0.26%)							
Household leader’s education (n = 2847)						<0.001 *		0.059		
	Kindergarten or none (a): 181 (6.36%)	162 (6.60%)	19 (4.88%)	26.77 (<0.001) * b differs from e **	Reference				Reference	0.0062
	Primary (b): 1710 (60.11%)	1437 (58.51%)	273 (70.18%)		1.61 (0.98, 2.65)	0.055	2.98 (1.28, 6.95)	0.011 *	0.6036	
	Lower Secondary (c): 342 (12.01%)	299 (12.17%)	43 (11.06%)		1.22 (0.69, 2.17)	0.485	1.80 (0.67, 4.87)	0.244		
	Upper Secondary (d): 370 (13.01%)	328 (13.36%)	42 (10.8%)		1.09 (0.61, 1.93)	0.764	2.76 (1.04, 7.33)	0.041 *		
	Higher (e): 242 (8.51%)	230 (9.36%)	12 (3.08%)		0.44 (0.21, 0.94)	0.034*	2.12 (0.65, 6.88)	0.213		
Education of Adolescent Mother (n = 2847)						<0.001 *		<0.001 *		
	Kindergarten or none (a): 14 (0.49%)	10 (0.41%)	4 (1.04%)	378.95 (<0.001) * b differs from c,d,e and c differs from d,e **	Reference				0.0051	0.0920
	Primary (b): 158 (5.55%)	81 (3.30%)	77 (19.79%)		2.37 (0.71, 7.89)	0.158	1.01 (0.25, 4.15)	0.987	Reference	
	Lower Secondary (c): 762 (26.78%)	561 (22.83%)	201 (51.67%)		0.89 (0.27, 2.88)	0.854	0.54 (0.13, 2.21)	0.389		
	Upper Secondary (d): 1690 (59.38%)	1589 (64.67%)	101 (25.96%)		0.15 (0.04, 0.51)	0.002 *	0.09 (0.02, 0.38)	0.001 *		
	Higher (e): 222 (7.80%)	216 (8.79%)	6 (1.54%)		0.06 (0.01, 0.28)	<0.001 *	0.01 (0.00, 0.15)	0.001 *		

Different pseudo R^2^: comparison of the full model (pseudo R^2^ = 0.4248) and the full model without an interesting factor; the full model adjusted for household poverty, residential area and region, HOH language, HOH religion, HOH education, education of the mother, parity, number of household members, number of children and adolescents in the household; * statistically significant at alpha = 0.05; ** statistically significant at Bonferroni correction alpha = 0.0005.

**Table 5 ijerph-19-13626-t005:** The proportion of women aged 20–24 years who were married or in union before age 15 years by socioeconomic status (SES), association between early marriage and socioeconomic status, and effect of each SES as illustrated by the PAF and different pseudo-R-squared values.

Category		Non-Early Marriage (0)	Early Marriage (1)	Chi^2^ (*p* Value)	Univariate		Multivariate		Population Attributable Fraction: PAF	Different Pseudo R^2^
No. (Column%)		No. (Column%)	No. (Column%)		OR (95%CI)	*p* Value	AOR (95%CI)	*p* Value		
Poverty status (n = 2953)						<0.001 *		0.772		
	Poorest (a): 648 (21.94%)	581 (21.12%)	67 (33.33%)	26.70 (<0.001) * a differs from e **	Reference				Reference	0.0020
	Poor (b): 692 (23.43%)	649 (23.58%)	43 (21.39%)		0.57 (0.38, 0.85)	0.006 *	0.94 (0.54, 1.62)	0.817	0.0181	
	Middle (c): 662 (22.42%)	616 (22.38%)	46 (22.89%)		0.64 (0.43, 0.95)	0.030 *	1.02 (0.57, 1.84)	0.947		
	Rich (d): 591 (20.02%)	553 (20.09%)	38 (18.91%)		0.59 (0.39, 0.90)	0.014 *	1.31 (0.71, 2.42)	0.393		
	Richest (e): 360 (12.19%)	353 (12.83%)	7 (3.48%)		0.17 (0.07, 0.37)	<0.001 *	0.77 (0.28, 2.09)	0.605		
Residential area (n = 2953)										
	Urban (municipal): 1171 (39.65%)	1088 (39.53%)	83 (41.29%)	0.24 (0.623)	Reference				0.1375	0.0033
	Rural (non-municipal): 1782 (60.35%)	1664 (60.47%)	118 (58.71%)		0.92 (0.69, 1.24)	0.623			Reference	
Residential region (n = 2953)						<0.001 *		0.022 *		
	Bangkok (a): 296 (10.02%)	281 (10.21%)	15 (7.46%)	20.83 (<0.001) * b differs from e **	Reference				Reference	0.0187
	Central (b): 848 (28.72%)	766 (27.83%)	82 (40.80%)		2.00 (1.13, 3.53)	0.016 *	1.72 (0.80, 3.70)	0.163	0.2143	
	Northern (c): 421 (14.26%)	389 (14.14%)	32 (15.92%)		1.54 (0.81, 2.89)	0.180	1.62 (0.68, 3.87)	0.273		
	Northeastern (d): 711 (24.08%)	667 (24.24%)	44 (21.89%)		1.23 (0.67, 2.25)	0.491	0.74 (0.32, 1.73)	0.489		
	Southern (e): 677 (22.93%)	649 (23.58%)	28 (13.93%)		0.80 (0.42, 1.53)	0.516	0.84 (0.33, 2.14)	0.719		
Household leader’s language (n = 2953)										
	Thai: 2538 (85.95%)	2355 (85.57%)	183 (91.04%)	4.64 (0.031) *	Reference				0.4929	0.0045
	Non-Thai: 415 (14.05%)	397 (14.43%)	18 (8.96%)		0.58 (0.35, 0.95)	0.033 *	0.45 (0.21, 0.96)	0.040 *	Reference	
Household leader’s religion (n = 2953)						0.118				
	Buddhism (a): 2511 (85.03%)	2328 (84.59%)	183 (91.04%)	6.20 (0.102)	Reference				Reference	0.0004
	Islam (b): 411 (13.92%)	394 (14.32%)	17 (8.46%)		0.54 (0.33, 0.91)	0.021			0.0009	
	Christianity (c): 29 (0.98%)	28 (1.02%)	1 (0.50%)		0.45 (0.06, 3.35)	0.440				
	Others (d): 0 (0.00%)	0 (0.00%)	0 (0.00%)							
	No religion (e): 2 (0.07%)	2 (0.07%)	0 (0.00%)							
Household leader’s education (n = 2949)						0.085				
	Kindergarten or none (a): 186 (6.31%)	173 (6.30%)	13 (6.47%)	8.49 (0.075)	Reference				Reference	0.0018
	Primary (b): 1676 (56.83%)	1550 (56.40%)	126 (62.69%)		1.08 (0.59, 1.95)	0.795			0.1320	
	Lower Secondary (c): 397 (13.46%)	366 (13.31%)	31 (15.42%)		1.12 (0.57, 2.20)	0.727				
	Upper Secondary (d): 384 (13.02%)	364 (13.25%)	20 (9.95%)		0.73 (0.35, 1.50)	0.395				
	Higher (e): 306 (10.38%)	295 (10.74%)	11 (5.47%)		0.49 (0.21, 1.13)	0.096				
Education of Adolescent Mother (n = 2952)						<0.001 *		<0.001 *		
	Kindergarten or none (a): 52 (1.76%)	46 (1.67%)	6 (2.99%)	143.16 (<0.001) * d differs from b,c and e differs from b,c,d **	Reference				0.0251	0.0661
	Primary (b): 235 (7.96%)	194 (7.05%)	41 (20.40%)		1.62 (0.64, 4.04)	0.301	0.84 (0.23, 3.07)	0.792	Reference	
	Lower Secondary (c): 780 (26.42%)	681 (24.75%)	99 (49.25%)		1.11 (0.46, 2.67)	0.808	0.43 (0.12, 1.57)	0.203		
	Upper Secondary (d): 1000 (33.88%)	952 (34.61%)	48 (23.88%)		0.38 (0.15, 0.94)	0.038 *	0.21 (0.06, 0.78)	0.019 *		
	Higher (e): 885 (29.98%)	878 (31.92%)	7 (3.48%)		0.06 (0.01, 0.18)	<0.001 *	0.02 (0.00, 0.12)	<0.001 *		

Different pseudo R^2^: comparison of the full model (pseudo R^2^ = 0.1762) and the full model without an interesting factor; the full model adjusted for household poverty, residential area and region, HOH language, HOH religion, HOH education, education of the mother, parity, number of household members, number of children and adolescents in the household; * statistically significant at alpha = 0.05; ** statistically significant at Bonferroni correction alpha = 0.0005.

## Data Availability

Not applicable.

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
