# Peer review of "Patterns of Socioeconomic Inequities in SDGs Relating to Children’s Well-Being in Thailand and Policy Implications"

_ijerph, 2022, doi:10.3390/ijerph192013626_

Round 1

Reviewer 1 Report

Waleewong and Yueayai: Patterns of socioeconomic inequities in SDGs relating to children’s well-being (Children Flourishing Index) in Thailand and policy implications 

The article uses survey data to explore socioeconomic inequities in several core health and wellbeing indicators, also selected as SDG indicators. Many of the indicators are not readily available in Thailand and therefore, specific surveys are needed to receive information on them. The data collection is not presented, but the response rates seem to be very high. Therefore, it is strange that the survey gives a weighted adolescent birth rate of 136.6 per 1000 women aged 15-19 years, when the referred data sources give rates between 51 and 54 per 1000. Does this discrepancy signal poor generalizability? 

The presentation of indicators is very general in methods, and the reader need to go supplement table to understand some of the indicators. I strongly recommend that the table is moved as an ordinary table. For those not familiar with the local circumstances, some of the indicators may need more explanation. One example is the “ethnic languages”, which are referred, but not clearly explained. 

Of the eight indicators, the authors have selected five ones for which Thailand does not benchmark very well. This is a good and well justified solution. 

The authors have calculated population-attributable fractions, but these are not systematically presented. It would be easier for the reader to present the selected indicators in a similar manner. 

I wonder if all requirements for using chi square test were fulfilled in all comparisons. 

It remained unclear if the authors have used all the available variables in the survey. For example, parity and/or family size could be important to adjust for. 

The addition of child flourishing index may not give any added value, since half of the indicators are not covered by this survey. The readers may have problems to compare Thailand’s score with the unweighted outcomes received from the survey. 

The main conclusion is that increased education differences decrease differences in the selected indicators. The other policy implication on increased decentralization is vaguer and should be elaborated in more detailed.

Add national data to the supplementary figure 1. 

The language and presentation require some revision, for example systematically give percentage after the units or to avoid sentences as “20 times lower”, which is quite inaccurate.

Author Response

Dear reviewers, 
Thank you so much for your comments. We edited our manuscript follow all reviewers and an academic editor suggestion, lots of changing but hopefully our manuscript will be better.
Best regards,
Khanuengnij yueayai and Orratai Waleewong

Author Response

(The authors gave the same response as above.)

Reviewer 3 Report

Dear authors,

Overall, I believe the work to have importance and I think it will be of use to other researchers.

However, is there an inconsistency between the use of the Pearson's chi-square test you mentioned in lines 98, 99 and 100 in your study and the chi-square p value given in the table?

Shouldn't there be a chi squared p-value in each row?

If the Pearson's chi-square test was performed for all variables, a post hoc chi-square test could be performed and the results could be indicated in symbols to show which variable caused the difference.

I hope to see a revised version in the future should you choose to make changes.

Author Response

(The authors gave the same response as above.)

Round 2

Reviewer 1 Report

Waleewong and Yueayai: Patterns of socioeconomic inequities in SDGs relating to children’s well-being (Children Flourishing Index) in Thailand and policy implications

The authors have made a great job in revising this article. After these changes, I can recommend that the article is accepted for publication.

While reading the paper, I notified some issues, which could be edited:

-          Line 30: many aspects of child’s – check the spaces.

-          Lines 92-93: consider giving only one decimal (this applies also for other parts of the article).

-          Lines 111-112: space after grade 1-6 and grade 7-9.

-          Line 134: test with Bonferroni correction instead of and.

-          Line 173: 2.61 instead of 22.61

-          Line 183: data were instead of data was

-          Discussion (no line numbers): …that the sample was not representative instead of not a good representative.

Author Response

Dear reviewer, 

Thank you for your comments. we edited following all your comments. However, for Line 173 the upper of 95%CI was 22.61. The range of 95%CI very wide due to small sample for no religion. 

Best regards,

Khanuengnij Yueayai

Reviewer 3 Report

Dear Authors,

I appreciate your great attention making the requested changes. 

Best wishes,

Round 3

Reviewer 1 Report

-